# A Linear Regression Approach for Best Scanline Determination in the Object to Image Space Transformation Using Pushbroom Images

**DOI:** 10.3390/s24175594

**Published:** 2024-08-29

**Authors:** Seyede Shahrzad Ahooei Nezhad, Mohammad Javad Valadan Zoej, Fahimeh Youssefi, Ebrahim Ghaderpour

**Affiliations:** 1Department of Photogrammetry and Remote Sensing, Faculty of Geodesy and Geomatics Engineering, K. N. Toosi University of Technology, Tehran 19967-15443, Iran; shahrzad.ahooei@email.kntu.ac.ir (S.S.A.N.); valadanzouj@kntu.ac.ir (M.J.V.Z.); youssefi@usx.edu.cn (F.Y.); 2Institute of Artificial Intelligence, Shaoxing University, 508 West Huancheng Road, Yuecheng District, Shaoxing 312000, China; 3Department of Earth Sciences and CERI Research Centre, Sapienza University of Rome, P.le Aldo Moro, 5, 00185 Rome, Italy; 4Earth and Space Inc., Calgary, AB T3A 5B1, Canada

**Keywords:** object-to-image transformation, pushbroom imagery, best scanline search/determination (BSS/BSD), linear regression model (LRM), machine learning, photogrammetry

## Abstract

The use of linear array pushbroom images presents a new challenge in photogrammetric applications when it comes to transforming object coordinates to image coordinates. To address this issue, the Best Scanline Search/Determination (BSS/BSD) field focuses on obtaining the Exterior Orientation Parameters (EOPs) of each individual scanline. Current solutions are often impractical for real-time tasks due to their high time requirements and complexities. This is because they are based on the Collinearity Equation (CE) in an iterative procedure for each ground point. This study aims to develop a novel BSD framework that does not need repetitive usage of the CE with a lower computational complexity. The Linear Regression Model (LRM) forms the basis of the proposed BSD approach and uses Simulated Control Points (SCOPs) and Simulated Check Points (SCPs). The proposed method is comprised of two main steps: the training phase and the test phase. The SCOPs are used to calculate the unknown parameters of the LR model during the training phase. Then, the SCPs are used to evaluate the accuracy and execution time of the method through the test phase. The evaluation of the proposed method was conducted using ten various pushbroom images, 5 million SCPs, and a limited number of SCOPs. The Root Mean Square Error (RMSE) was found to be in the order of ten to the power of negative nine (pixel), indicating very high accuracy. Furthermore, the proposed approach is more robust than the previous well-known BSS/BSD methods when handling various pushbroom images, making it suitable for practical and real-time applications due to its high speed, which only requires 2–3 s of time.

## 1. Introduction

Over the past few decades, linear pushbroom cameras have become increasingly popular in various applications of remote sensing and photogrammetry [1], including global and topographic mapping, environmental monitoring, change detection, geological survey, target detection, and three-dimensional (3D) reconstruction [2,3]. Linear array pushbroom cameras can provide high-resolution panchromatic or multispectral satellite images with a wider field of view and high revisit frequency, making it easier to map at different scales [4,5].

However, processing linear pushbroom images is more complex than processing frame-type images [5,6,7]. In linear pushbroom imaging sensors, the two-dimensional (2D) image scene is the result of capturing sequential one-dimensional (1D) scanlines at different instants in time, each of which has distinct Exterior Orientation Parameters (EOPs) [8,9]. This dynamic imaging geometry complicates the process of transforming 3D ground space to 2D image space [10,11]. Unlike frame-type images, where each image has one set of EOPs, the position and orientation of the perspective center varies line-by-line when using pushbroom images [12,13]. This variation in EOPs for each scanline makes it challenging to determine the exact time of exposure required to achieve EOPs for each ground point. This problem is called the Best Scanline Search/Determination (BSS/BSD) and often known as determining the precise time of exposure [14,15]. The EOPs of the corresponding scanline are required for Collinearity Equation- (CE) based object-to-image transformation, which is unavailable for pushbroom images [16,17,18]. This transformation process is fundamental in the geometric processing of linear pushbroom images, including DTM generation [19], image matching [19], epipolar resampling [20], stereoscopic measurements, and orthorectification [21,22].

A great deal of research has been undertaken on the topic of object-to-image space transformation using pushbroom images, known as the Best Scanline Search (BSS). Several methods have been proposed to solve this challenge, including the Sequential Search (SS), the Bisecting Window Search (BWS) [23], the Newton Raphson (NR) [24], the Central Perspective Plane (CPP) [25], the General Distance Prediction (GDP) [19], the Optimal Global Polynomial (OGP) using the Genetic Algorithm (GA) [21], and the Artificial Neural Network (ANN) [21].

In the SS, each ground point is back projected from the ground/object space to the image space for the total number of image scanlines, and the value of the first component of the CE is evaluated, such that if this value approaches zero or close to zero, the corresponding scanline is deemed as the best scanline. The BWS has effectively diminished the search space of the SS by successively halving the image space and incorporating the first and last scanlines of the image in the assessment of the CE’s first component during an iterative procedure until the best scanline is attained. The NR has been employed to ascertain the root of the initial component of the CE, in accordance with the inherent characteristics of linear array pushbroom images. Furthermore, this methodology necessitates the application of the CE for each ground point throughout the iterative process. The CPP is introduced for linear array aerial images, endeavoring to address the challenge of identifying the best scanline by delineating the CPP corresponding to each ground point through iterative process of geometric equations. Additionally, in the final stage, the CE is utilized within the iterative process in accuracy improvement stage, albeit with a reduced number of iterations. The GDP approach is based on the back-projection of each ground point to image space through the CE by employing EOPs of the first scanline. Subsequently, the GDP, which quantifies the spatial interval between the projected image point and the first line of the image, is computed and employed for updating value via specific mathematical formulations. This iterative process persists until the predetermined stopping criterion is satisfied, leading to the identification of the best scanline corresponding to the ground point using linear interpolation. This approach operates through an iterative framework involving multiple interpolations, thereby necessitating a moderate number of computational resources. The determination of the best scanline within the framework of the OGP and ANN is conducted through a three-step procedure. Initially, the BSS is executed by utilizing many simulated points, thereby computing the estimated scanline value. Subsequently, the ground points transferred from the object space to the image space using estimated scanline value, and their accuracy are refined within the image space. Following this, a model is established to relate the final scanline values calculated for the aforementioned points in conjunction with the computed approximate value, so that it can facilitate the application of this model to other points in the third step, devoid of the necessity to transfer between the two spaces and independent of the CE. In the second stage, both GA and ANN have been employed separately.

However, all these methods except the OGP and ANN require consecutive complicated iterations based on the CE with high computational cost, making them unsuitable for practical applications due to low efficiency and high time consumption. Two non-iterative BSD methods were proposed by the authors of [21] who suggested two distinct BSD approaches regardless of using the CE. Although these methods provide proper accuracy in less time compared to the previous methods, their accuracy and execution time will be highly dependent on the GA and ANN parameters and procedure, and they also require a number of control points called Simulated Control Points (SCOPs) in the training phase.

As the current solutions to solve BSS/BSD problems are still computationally demanding, time-consuming, and not feasible for real-time tasks, there is a need for faster, non-iterative, and simpler BSS/BSD methods. Since Machine Learning (ML) methods have the advantage of simplicity in implementation and higher accuracy [26], this paper proposes a non-iterative, innovative, and fast BSD approach by taking a new look at the regression model as one of the ML methods. This approach removes the necessity for iterative search and the use of the CE in the BSD process, significantly reducing the time required. Previous methods often involved iterative procedures or the need to use the CE, resulting in high time consumption in the object-to-image space transformation process. In contrast, the proposed single-step LRM offers high speed and requires very little time. The proposed method involves two main steps: the training step and the test step. In the training phase, a few SCOPs are used to obtain the regression model parameters. In the test phase, the proposed method performs the BSD procedure of the other set of Simulated Check Points (SCPs) through the defined regression model without any iteration of the CE, resulting in a very short processing time. Ten pushbroom satellite images were employed in the proposed method, and the results were used in performance evaluation. Furthermore, the proposed method has been compared with some previous methods such as the BWS, NR, OGP, and ANN.

The rest of this paper is organized as follows. The relevant mathematical models, dataset description, and explanation of the proposed method are included in Section 2. The experimental results and accuracy assessments are provided in Section 3. Further discussion is given in Section 4. Finally, the conclusions are outlined in Section 5.

## 2. Materials and Methods

This section is dedicated to the explanation of the used datasets, the theoretical foundations and the description of the proposed method. Figure 1 displays the proposed BSD method’s flowchart. The proposed method based on a regression model has two main steps. The first stage is the training stage of the model, which is performed by employing SCOPs, and the second stage includes the evaluation of BSD using SCPs. Section 2.3 will provide additional explanations about the SCOPs and SCPs generation as well as the proposed method description. An overview of the proposed method is given in Table 1 for clarification.

### 2.1. Dataset Description

As reported in Table 2, the experimental datasets include ten different satellite pushbroom images acquired by various sensors including IKONOS, Pleiades 1A and 1B, QuickBird, SPOT6, SPOT7, WorldView 1, and WorldView 2. A comprehensive selection of ten images was made, illustrating a variety of sensor characteristics, including dimensions and spatial resolutions, alongside disparate land cover and topographical conditions, such as urban areas, flat terrains, agricultural regions, and combinations thereof, for a more thorough assessment of the proposed BSD (see Figure 2). Specifically, the spatial resolution of these images exhibited a range from 0.5 to 6 m. It is noteworthy that all images utilized in this research were supplemented by Rational Polynomial Coefficients (RPCs) files; the necessary Ground Control Points (GCPs) were derived from these supplementary files. Furthermore, elevation data for the GCPs as well as the average height of the study areas were derived from the accessible Digital Elevation Model (DEM) resources.

### 2.2. The Collinearity Equation (CE)

From the photogrammetric point of view, the relationship between the 2D image space and the 3D ground space is established using mathematical models [21,27]. The CE is a well-known mathematical model based on the geometry of the image at the time of imaging, which has been widely used for pushbroom images [4,28]. In the case of pushbroom images, the extended CE can be expressed as Equation (1).
(1)$$x=-f\frac{r_{11}^{i}\left(X-X_{S}^{i}\right)+r_{12}^{i}\left(Y-Y_{S}^{i}\right)+r_{13}^{i}\left(Z-Z_{S}^{i}\right)}{r_{31}^{i}\left(X-X_{S}^{i}\right)+r_{32}^{i}\left(Y-Y_{S}^{i}\right)+r_{33}^{i}\left(Z-Z_{S}^{i}\right)}=0\;y=-f\frac{r_{21}^{i}\left(X-X_{S}^{i}\right)+r_{22}^{i}\left(Y-Y_{S}^{i}\right)+r_{23}^{i}\left(Z-Z_{S}^{i}\right)}{r_{31}^{i}\left(X-X_{S}^{i}\right)+r_{32}^{i}\left(Y-Y_{S}^{i}\right)+r_{33}^{i}\left(Z-Z_{S}^{i}\right)}$$
where (x, y) are 2D image coordinates of an arbitrary point, f is the focal length, (X, Y, Z) are 3D ground coordinates of an arbitrary point, *i* refers to the scanline number, and ($X_{S}^{i},Y_{S}^{i},\;Z_{S}^{i})$ and ($r_{11}^{i},\ldots ,\;r_{33}^{i}$) are the positions of the perspective center in object space and rotation matrix elements of rotation angles ω, φ and κ (EOPs), respectively.

According to Equation (1), to use the CE, reliable EOPs are necessary. Therefore, a space resection method is required to obtain the EOPs of all scanlines, considering the dynamic nature of pushbroom images. The Multiple Projection Center (MPC) model was used in the space resection step in this study. The equations for the MPC model equations can be found in Equation (2) [12,29].
(2)$${X_{s}}^{i}\left(t\right)=X_{0}+X_{1}\left(t_{i}\right)+X_{2}(t_{i}^{2})\;{Y_{s}}^{i}\left(t\right)=Y_{0}+Y_{1}\left(t_{i}\right)+Y_{2}(t_{i}^{2})\;{Z_{s}}^{i}\left(t\right)=Z_{0}+Z_{1}\left(t_{i}\right)+Z_{2}(t_{i}^{2})\;{w_{s}}^{i}\left(t\right)=w_{0}+w_{1}\left(t_{i}\right)+w_{2}(t_{i}^{2})\;{\varphi_{s}}^{i}\left(t\right)=\varphi_{0}+\varphi_{1}\left(t_{i}\right)+\varphi_{2}(t_{i}^{2})\;{k_{s}}^{i}\left(t\right)=k_{0}+k_{1}\left(t_{i}\right)+k_{2}(t_{i}^{2})$$
where $t_{i}$ is the *i*-th scanline’s exposure time (equivalent to the satellites’ along-track coordinate); $X_{0}$, $X_{1}$, …, and $k_{2}$ are the reference scanline EOPs that were determined during the space resection phase; ${X_{s}}^{i}\left(t\right)$,${Y_{s}}^{i}\left(t\right)$, …, and ${k_{s}}^{i}\left(t\right)$ are the *i*-th scanline’s EOPs.

### 2.3. Proposed Method

#### 2.3.1. SCOPs and SCPs Generation

As previously mentioned, the proposed method comprises two key phases—a training step and a testing step—utilizing two distinct groups of points: Simulated Control Points (SCOPs) and Simulated Check Points (SCPs). While both sets are generated in a similar manner, SCOPs are utilized during the training phase with a limited number of points, whereas SCPs are employed in the testing phase with a substantially larger quantity of points. To generate the points, regular grids are created separately in the image space for both SCOPs and SCPs, giving them image coordinates based on where they are placed in the grid. Then, the image-simulated points are transferred to the object space using the CE, the study area’s average height (determined from real GCPs), and the EOPs of all scanlines obtained through the MPC model. This results in sets of SCOPs and SCPs with known image coordinates as well as object coordinates, which are ready for processing and the evaluation of BSD.

#### 2.3.2. Linear Regression

One of the simplest and most widely used methods among statistical and machine learning algorithms is the linear regression model [30,31]. Linear regression is a method used to establish the linear relationship between dependent and independent variables [32,33].

In regression models, the independent variables predict the dependent variables [31]. The regression model with a single independent variable is known as Simple Linear Regression (SLR) [32]. The formula for SLR is given by Equation (3):(3)$$y=a_{0}+a_{1}x\;$$
where *y* is the dependent variable, *x* is the independent variable, $a_{0}$ and $a_{1}$ are the intercept and linear term values, respectively.

The goal of Multivariate Linear Regression (MLR) is to model the linear relation between one or more independent variables and a dependent variable [32]. The formula for simple MLR is given by Equation (4):(4)$$y_{i}=a_{0}+a_{1}x_{i1}+\cdots +a_{m}x_{im},\;\;\;\;\;\;\;\;\;\;1\leq i\leq n$$
Y=y1y2⋮yn,  A=a0a1⋮am,  X=1x11⋯x1m1x21⋯x2m⋮⋮⋮⋮1xn1⋯xnm
where $y_{i}$ is the single dependent variable, ($a_{0}$,…,${\;a}_{m}$) are the coefficients forming the A matrix, and $x_{ij}$ are the independent variables forming the X matrix.

Finally, the least square method (LSM), Equation (5) is used to find the best line or curve that fits the data sets, minimizing the cumulative squared residual errors [34,35].
(5)$$A{=(X^{T}X)}^{-1}X^{T}Y$$

#### 2.3.3. Polynomial Regression Model (PRM)

Polynomial regression is a type of multiple regression that involves modeling with an $n_{th}$ degree polynomial. This regression is used when there is a curvilinear relationship between the independent and dependent variables [36]. The general model for PR is represented by Equation (6) [37].
(6)$$y=a_{0}+a_{1}x+a_{2}x^{2}+\cdots +a_{m}x^{n}$$

In Equation (6), y represents the dependent variable, x represents independent variable, ($a_{0}$, …,$\;a_{m}$) are the regression coefficients of the independent variable, and *n* represents the polynomial degree or order of the regression model. If there are multiple independent variables, Equation (6) can be rewritten using the **X** matrix, like Equation (4), but with an $n_{th}$ degree polynomial.

#### 2.3.4. Linear Regression BSD Model

The Linear Regression Model (LRM)–BSD is divided into two steps, namely the training step and the test step. In the training step, the SCOPs with known ground and image coordinates are used to obtain the model parameters. The SCPs are then used for every unknown ground point for accuracy assessment. During the Least Squares Method (LSM), the unknown parameters of the MLR model are calculated using SCOPs. The MLR model takes the ground coordinates (*X*, *Y*) as input parameters and the row number (*r*) as the target parameter. This way, the transformation between ground space and image space is established. In the test step, the MLR model is applied to the ground coordinate (*X*, *Y*) of SCPs, and the row values are estimated. Finally, the accuracy assessment of SCPs is performed using a Root Mean Square Error (RMSE) value. The PRM is also applied to the proposed method for a more accurate evaluation using SCOPs and SCPs. The two steps and the accuracy assessment are like the case of using the MLR model.

#### 2.3.5. Accuracy Assessment

In evaluating the efficiency of the proposed method both in comparison with prior research and with the effective parameters of the current method, several measurement criteria have been employed. These criteria include the Root Mean Square Error (RMSE), dr_max_ (the maximum BSD error among all SCPs), and execution time. The RMSE and dr_max_ values are acquired by comparing the best scanline value derived from the proposed LRM–BSD approach with the exact scanline value provided during the simulation stage. Additionally, the number of SCOPs and suitability of the models are examined to obtain a robust model.

## 3. Results

The proposed BSD method’s performance is assessed for efficiency in both LRM and PRM modes. The model structure is clear, and the number of unknown parameters is known, allowing for an examination of the required SCOPs to solve the model and achieve desired accuracy. The number of SCOPs considered are ten, thirty, fifty, and one hundred per image, with five million SCPs employed for each image. Results are compared with other methods, such as the NR, BWS, ANN, and OGP using quantitative measures including RMSE, computation time, dr_max_, and the number of required SCOPs for comparison.

Experimental results are achieved using Intel Core i7 hardware with a 2.90 GHz processor, HD graphics 620, and 8 GB RAM. The analysis of changes in the number of SCOPs is performed by applying four groups of SCOPs. A numerous number of SCPs provides a more accurate assessment of the proposed BSD method’s robustness. According to the results presented in Table 3, increasing the number of SCOPs has a negligible effect on ultimate accuracy in terms of the RMSE and dr_max_. Interestingly, no significant difference is found in the computational time between these four groups (around 2 s). This issue reveals that the limitation in obtaining, and the number of SCOPs does not interfere with the proposed BSD, and it achieves very high accuracy (the RMSE is equal to ten to the negative power of nine) with only a small number of SCOPs (up to 50) in a very short amount of time.

The SPI image includes urban areas with dense buildings, and this issue can be the reason for its lower accuracy compared to other images (an order of accuracy reduction in the RMSE). Furthermore, taking into account that the CS image, possessing a resolution of 6 m, represents the lowest resolution within the used dataset, the obtained RMSE of this image reached 10 to the power of negative 10. Thus, despite using images with different characteristics, there is no correlation between the number of SCOPs, land cover, spatial resolution of the images, and final accuracy. This confirms the success of the proposed BSD using a variety of satellite images from all aspects.

The comparison results of the proposed BSD with other methods like the NR, BWS ANN–BSD, and OGP–BSD are reported in Table 4. Quantitative metrics including the RMSE, computation time, dr_max_, and the number of SCOPs are used to conduct the comparison. The results of this comparison are also illustrated in Figure 3, Figure 4 and Figure 5 for better comprehension.

The proposed method requires significantly less time than all the other methods. The NR and BWS require 250 and 750 times more computational time than the proposed method, respectively. The reason for this significant reduction in time is the non-iterative procedure used in the proposed method, regardless of using the CE. On the other hand, the NR and BWS require a lot of CE iterations for each ground point, and their accuracy depends on this. However, the BWS cannot achieve sub-pixel accuracy, which is essential for photogrammetric applications since the dr_max_ value for this method is one pixel. Additionally, the BWS takes the most computation time due to the large search space and more iteration of the CE required.

Furthermore, the discrepancies between the RMSE and dr_max_ of the proposed method and NR are negligible, (Both have the RMSEs and dr_max_ values of the order of 10 to the power of 9 or 10). The comparison of the ANN–BSD and OGP–BSD with the proposed method reveals that the LRM achieves a higher RMSE by using fewer SCOPs in less time. The main reason for this difference is that the ANN–BSD and OGP–BSD require more SCOPs than the LRM in the error modeling phase. The number and distribution of these SCOPs directly affect the final accuracy of modeling and computation time to ensure accuracy in determining the best scanline.

According to Figure 3, Figure 4 and Figure 5 among the compared methods, it can be observed that the NR and LRM exhibit superior accuracy, characterized by the RMSEs approximately zero. Subsequently, the OGP and ANN present the RMSEs of about 0.3 pixels. The BWS demonstrates the greatest level of RMSE, with a recorded value of 0.6 pixels. Furthermore, the BWS exhibits the most considerable computational time, amounting to 1300 s, due to the necessity of more repetition of the CE for each ground point across all images. The NR follows closely in terms of time consumption, requiring 500 s, which is the second longest duration after BWS. This issue can also be ascribed to the iterative application of the CE; however, it is noteworthy that the frequency of repetitions for the NR is lower than that of the BWS. In contrast, the OGP demonstrates an average execution time of merely 7 s, while both ANN and LRM exhibit equivalent processing times ranging from 2 to 3 s. It is essential to highlight that the RMSE of the LRM is very close to zero, whereas the RMSE of the ANN is approximately 0.3 pixels. Additionally, the behavior of the dr_max_ can be inferred from the RMSE values. Consequently, the BWS, characterized by the dr_max_ equivalent to one pixel, exhibits the maximum error in comparison to the other methods. The LRM and NR, both exhibiting the dr_max_ values of approximately zero, reside at the lower boundary of the graph, while the ANN and OGP occupy a mid-range position among the methodologies, achieving sub-pixel accuracy within the interval of 0.5 to 0.7 pixels. Based on this evidence, the proposed LRM is preferable to previous studies for real-time applications due to its high accuracy, low required time, and minimum dr_max_.

## 4. Discussion

The task of searching/determining the best scanline for mapping the object space onto the image space is a key issue when dealing with pushbroom images. Well-known existing research in this domain commonly relies on the principle of the CE, which often leads to significant computational demands due to its intricate nature. In response to this challenge, a novel approach has been introduced to reduce the computational process, minimize time requirements, and eliminate the necessity for the CE.

The accuracy and time-consumption of the BSS/BSD method directly affect the accuracy and total time required for photogrammetric products. So, the proposed method’s efficiency can be evaluated based on these two factors. As expected, the experiments demonstrated that the LRM–BSD achieved sub-pixel accuracy in the shortest possible time compared to previous studies. A notable decrease in time is attributed to the omission of the individual usage of the CE for each ground point. In the previous approaches, BSS methods like NR and BWS the transformation of ground coordinates to image coordinates necessitated the application of the CE through an iterative procedure, involving a search in the image space for each projected ground points. Conversely, the proposed method eliminates the dependence on the CE by determining the best scanline for each ground point through an LRM regardless of searching in iterative manner. The ANN and OGP do not depend on the CE; nevertheless, the LRM–BSD is considered more advantageous due to its improved precision, efficiency, and minimum number of SCOPs requirements. The RMSE achieved by the LRM method ($10^{-9}\approx 0$) is suitable for photogrammetric applications, such as orthorectification with low computation time for many points.

Considering that SCOPs are pivotal for the training and computation of LRM parameters, the distribution and precision of these points substantially impact the final accuracy of the proposed algorithm making use of SCPs. In the simulation phase of SCOPs and SCPs, the sole source of error arises from inaccuracies in the specification of EOPs. Given that these parameters have been derived and approximated utilizing real GCPs through the MPC model, any discrepancies in the GCPs will consequently spread to the determination of the EOPs. Furthermore, the LRM inherently possesses a degree of error and uncertainty, which can be acknowledged as a potential source of methodological errors. Nonetheless, based on the achieved RMSE and dr_max_, the influence of above-mentioned error sources on the final accuracy can be deemed negligible.

As mentioned earlier, providing SCOPs characterized by suitable distribution across each image and precision will lead to robust accuracy of the proposed method. Keeping this point in mind regarding the inputs of the LRM, the level of uncertainty associated with the outputs was assessed in terms of two key parameters: dr_max_ and the RMSE (overall accuracy). Based on the quantity of results derived from all images pertaining to these two parameters, the proposed method exhibits a minimal degree of uncertainty, which is deemed advantageous. Thus, due to the improved accuracy metrics presented by the LRM, compared to the previous studies, this method is superior for application in photogrammetric contexts.

The space resection phase to obtain EOPs preceded the LRM–BSD, but if EOPs are available from other sources, the space resection step can be omitted. Therefore, the proposed LRM–BSD approach can be used even in the absence of EOPs. This issue emphasizes the algorithm’s potential to perform independently of presumptions or obligatory parameters. Nonetheless, it is imperative to establish real GCPs in instances where EOPs are unavailable, in addition to preparing a finite quantity of SCOPs for the training phase. Moreover, the proposed LRM–BSD requires fewer SCOPs than the ANN and OGP BSD due to the fixed and certain unknown parameters of the model (three unknown parameters based on the LRM structure). Although there were no significant differences in computation time and the RMSE found by increasing the number of SCOPs, it is preferable to use a smaller number of them due to the limitation in providing SCOPs.

Further investigation was undertaken using the PRM–BSD, and the results obtained from this analysis had the same accuracy as the LRM–BSD. The coefficients with degrees higher than linearity had values close to zero, indicating the adequacy of the LRM in solving the BSD problem.

## 5. Conclusions

This paper presents an approach to object-to-image transformation using satellite pushbroom images through the LRM in BSD. This method is based on the LRM, comprising of training and test phases involving SCOPs and SCPs respectively. It is easy to implement and not dependent on using the CE iteratively. The accuracy assessment was carried out using ten, thirty, fifty, and one hundred groups of SCOPs along with five million SCPs. Additionally, the LRM–BSD was compared to other methods, such as the BWS, NR, ANN, and OGP methods based on evaluation metrics, such as the RMSE, dr_max_, and execution time. The experimental results indicate that the proposed LRM–BSD procedure is faster than the other methods and has a better sub-pixel accuracy. The LRM is 250 times faster than the NR, but both methods have the same level of accuracy with the RMSEs very close to zero (pixel). The BWS is not capable of obtaining accuracies better than 0.5 pixels based on the obtained dr_max_ and requires significant execution time, on average of 1300 s. On the other hand, the LRM obtains much better accuracies than the ANN and OGP (the RMSE values ranged between 0.28 and 0.32 pixels) in a shorter time and with fewer required SCOPs. The required time of these two methods is between 3 and 8 s, and the number of required SCOPs is 500. However, the LRM takes 2 s on average, and the maximum number of needed SCOPs is 50. The proposed LRM has potential applications in orthophoto generation, epipolar resampling, and other similar tasks. Therefore, future research will focus on applying the LRM to these tasks. It is suggested that the LRM on different aerial pushbroom images be applied in future research to further confirm the findings.

## Figures and Tables

**Figure 1 sensors-24-05594-f001:**
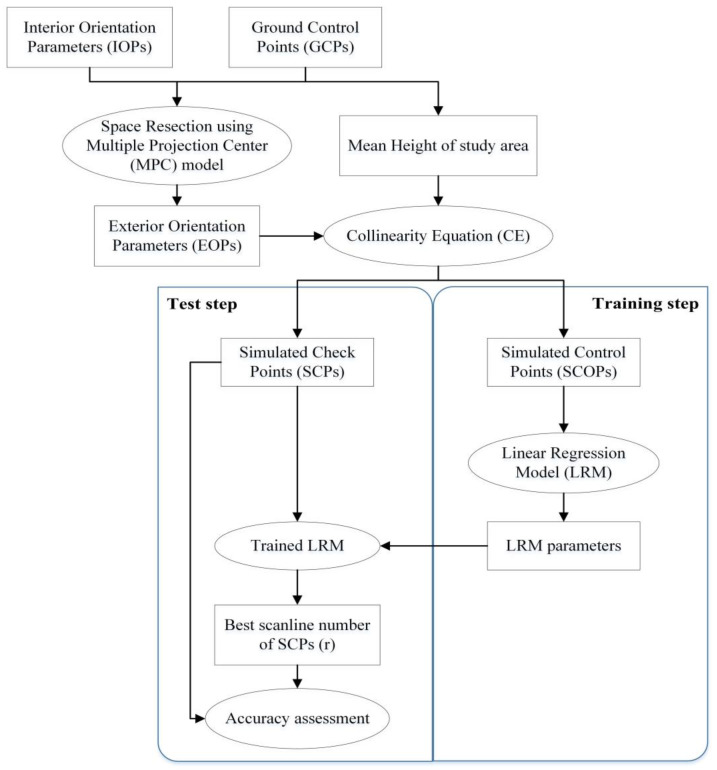
The proposed BSD approach’s flowchart.

**Figure 2 sensors-24-05594-f002:**
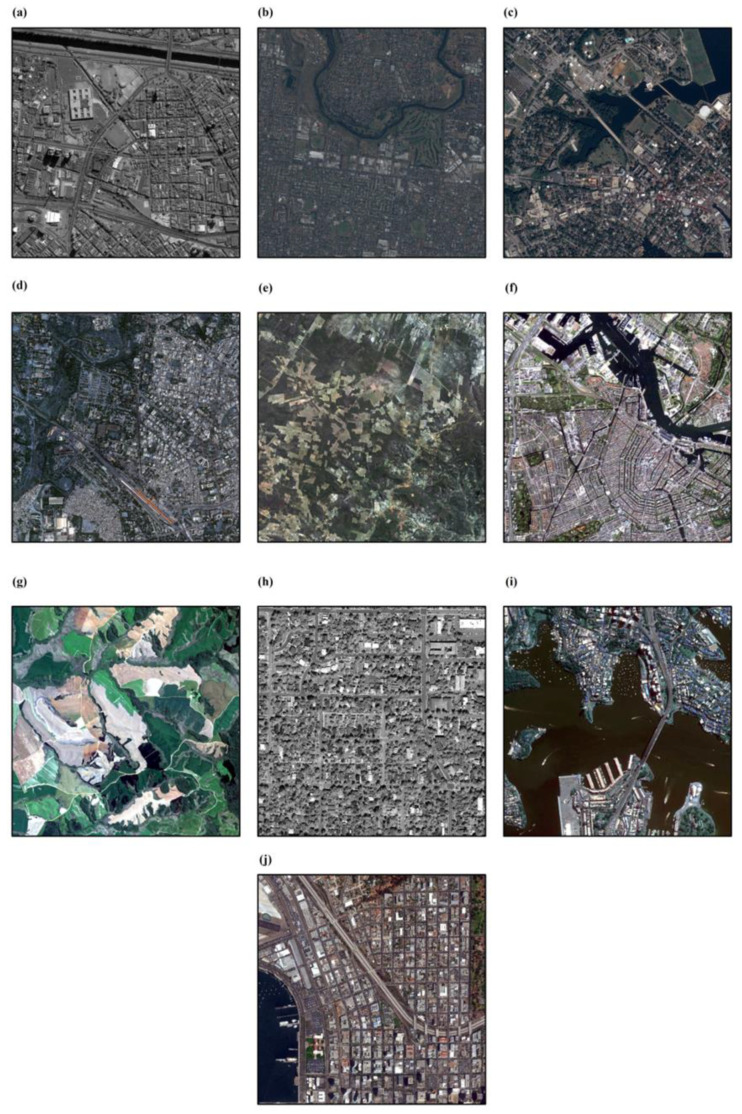
Employed images with different coverage area: (**a**) SPI, (**b**) MP, (**c**) AP, (**d**) JQB, (**e**) JS, (**f**) AS, (**g**) CS, (**h**) BWV, (**i**) SWV, and (**j**) SDWV.

**Figure 3 sensors-24-05594-f003:**
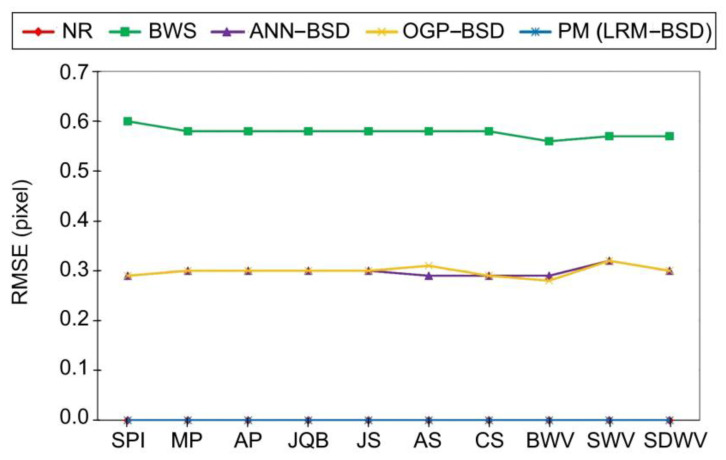
The comparison of the proposed BSD with previous studies in terms of the RMSE. The NR and PM values are very close to zero.

**Figure 4 sensors-24-05594-f004:**
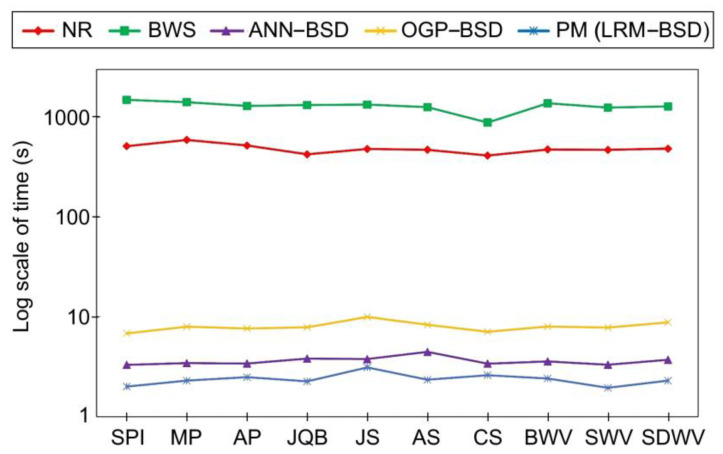
Comparison of the proposed BSD with previous studies in terms of time.

**Figure 5 sensors-24-05594-f005:**
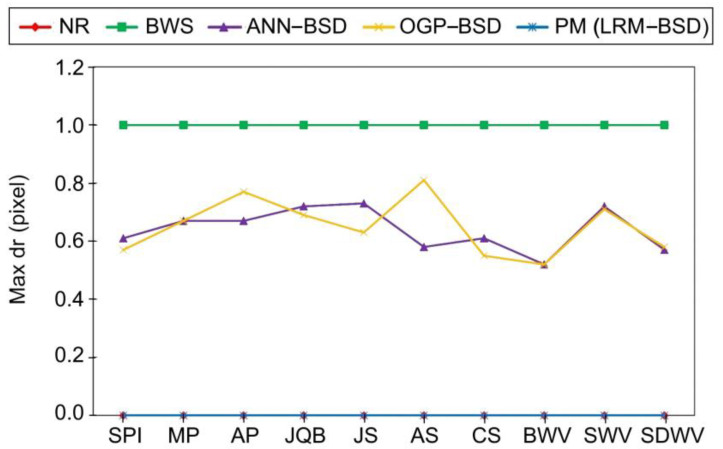
The comparison of the proposed BSD with previous studies in terms of dr_max_. The NR and PM values are very close to zero.

**Table 1 sensors-24-05594-t001:** An overview of the proposed method.

		Mathematical Model	Inputs	Outputs
Preprocessing steps	Space resection	MPC model	IOPs Image coordinates of GCPs Ground coordinates of GCPs	EOPs of the scanlines for the whole image
	SCOPs and SCPs generation	The CE	IOPs EOPs Mean height of the study area	Image and ground coordinates of SCOPs Image and ground coordinates of SCPs
LRM–BSD steps	Training phase	LRM model	Image coordinates of SCOPs Ground coordinates of SCOPs	The fitted LRM model for object-to-image transformation
	Testing phase	LRM model	Ground coordinates of SCPs	Image coordinates of SCPs predicted through the fitted LRM model Accuracy assessment (RMSE, time, dr_max)_

**Table 2 sensors-24-05594-t002:** Specifications of the experimental datasets.

Dataset	Imaging Sensor	Coverage Zone	Spatial Resolution (m)	Subset Size (Pixel)
SPI	IKONOS	Sao Paulo, Brazil	0.80	8300 × 8600
MP	Pleiades 1A	Melbourne, Australia	0.5	6000 × 7000
AP	Pleiades 1B	Annapolis, MD, USA	0.5	6057 × 5636
JQB	QuickBird	Jaipur, India	0.6	6000 × 6000
JS	SPOT 6	Jaicos, Brazil	1.5	6200 × 6600
AS	SPOT 7	Amsterdam, The Netherlands	1.5	5824 × 6616
CS	SPOT 7	Curitiba, Brazil	6	2597 × 1463
BWV	WorldView 1	Boulder, CO, USA	0.5	6000 × 6000
SWV	WorldView 2	Sydney, Australia	0.5	6000 × 6000
SDWV	WorldView 2	San Diego, CA, USA	0.5	3996 × 4015

**Table 3 sensors-24-05594-t003:** The results of the proposed BSD by testing different number of SCOPs.

Image	# of SCOPs	RMSE	Time	dr_max_
SPI	10	4.5 × 10^−9^	2.72	7.4 × 10^−9^
	30	4.0 × 10^−9^	2.37	7.6 × 10^−9^
	50	1.2 × 10^−9^	2.00	2.7 × 10^−9^
	100	4.0 × 10^−9^	2.02	7.2 × 10^−9^
MP	10	3.1 × 10^−9^	2.10	6.4 × 10^−9^
	30	2.3 × 10^−9^	2.29	3.3 × 10^−9^
	50	2.3 × 10^−9^	2.13	4.4 × 10^−9^
	100	4.8 × 10^−9^	2.25	9.0 × 10^−9^
AP	10	1.5 × 10^−9^	2.48	4.0 × 10^−9^
	30	2.1 × 10^−9^	2.25	4.5 × 10^−9^
	50	2.9 × 10^−9^	2.40	5.8 × 10^−9^
	100	1.7 × 10^−9^	2.68	3.1 × 10^−9^
JQB	10	7.0 × 10^−10^	2.52	1.6 × 10^−9^
	30	4.4 × 10^−10^	2.25	9.5 × 10^−10^
	50	5.9 × 10^−10^	3.16	1.0 × 10^−9^
	100	2.6 × 10^−9^	2.75	3.9 × 10^−9^
JS	10	8.8 × 10^−10^	3.11	2.4 × 10^−9^
	30	2.0 × 10^−9^	2.60	4.4 × 10^−9^
	50	9.9 × 10^−10^	2.28	3.3 × 10^−9^
	100	2.1 × 10^−9^	2.63	3.3 × 10^−9^
AS	10	5.7 × 10^−10^	2.49	1.0 × 10^−9^
	30	3.1 × 10^−10^	2.31	7.6 × 10^−10^
	50	1.8 × 10^−10^	2.34	3.9 × 10^−10^
	100	1.6 × 10^−9^	2.82	2.1 × 10^−9^
CS	10	1.7 × 10^−10^	2.60	3.7 × 10^−10^
	30	2.8 × 10^−10^	2.36	5.5 × 10^−10^
	50	2.8 × 10^−10^	2.42	5.5 × 10^−10^
	100	6.0 × 10^−10^	2.09	1.2 × 10^−9^
BWV	10	1.8 × 10^−9^	2.30	4.3 × 10^−9^
	30	9.8 × 10^−10^	2.41	2.5 × 10^−9^
	50	3.3 × 10^−9^	2.38	5.5 × 10^−9^
	100	3.9 × 10^−9^	2.50	7.2 × 10^−9^
SWV	10	9.0 × 10^−10^	1.94	1.7 × 10^−9^
	30	3.2 × 10^−9^	1.95	4.1 × 10^−9^
	50	3.9 × 10^−9^	1.79	7.3 × 10^−9^
	100	3.0 × 10^−9^	1.93	4.3 × 10^−9^
SDWV	10	1.0 × 10^−9^	2.29	1.7 × 10^−9^
	30	2.6 × 10^−9^	1.95	4.5 × 10^−9^
	50	1.1 × 10^−9^	1.95	1.8 × 10^−9^
	100	1.4 × 10^−9^	2.03	2.1 × 10^−9^

**Table 4 sensors-24-05594-t004:** The results of the proposed BSD compared to some previous studies.

Dataset	Measurement Criteria	Method				
		Newton Raphson (NR) [24]	Bisecting Window Search (BWS) [23]	ANN–BSD [21]	OGP–BSD [21]	Proposed Method (LRM)
SPI	RMSE (pixel)	5.9 × 10^−10^	0.60	0.28	0.28	1.2 × 10^−9^
	Time (second)	512.00	1491.20	3.30	6.80	2.00
	dr_max_ (pixel)	1.73 × 10^−9^	1	0.61	0.57	2.7 × 10^−9^
	Number of SCOPs	-	-	400	400	50
MP	RMSE (pixel)	1.0 × 10^−9^	0.58	0.30	0.30	2.3 × 10^−9^
	Time (second)	591.70	1415.10	3.43	7.94	2.29
	dr_max_ (pixel)	2.6 × 10^−9^	1	0.67	0.67	3.3 × 10^−9^
	Number of SCOPs	-	-	500	500	30
AP	RMSE (pixel)	9.6 × 10^−10^	0.58	0.30	0.30	1.5 × 10^−9^
	Time (second)	520.21	1396.87	3.40	7.64	2.48
	dr_max_ (pixel)	1.2 × 10^−9^	1	0.67	0.77	4.0 × 10^−9^
	Number of SCOPs	-	-	500	500	10
JQB	RMSE (pixel)	4.4 × 10^−10^	0.58	0.30	0.30	4.4 × 10^−10^
	Time (second)	425.94	1324.22	3.81	7.84	2.25
	dr_max_ (pixel)	1.2 × 10^−9^	1	0.72	0.69	9.5 × 10^−10^
	Number of SCOPs	-	-	500	500	30
JS	RMSE (pixel)	6.2 × 10^−10^	0.58	0.30	0.30	8.8 × 10^−10^
	Time (second)	484.48	1355.63	3.77	9.96	3.11
	dr_max_ (pixel)	2.2 × 10^−9^	1	0.73	0.63	2.4 × 10^−9^
	Number of SCOPs	-	-	500	1000	10
AS	RMSE (pixel)	2.5 × 10^−10^	0.58	0.29	0.31	1.8 × 10^−10^
	Time (second)	460.37	1270.50	4.45	8.31	2.34
	dr_max_ (pixel)	6.2 × 10^−10^	1	0.58	0.81	3.9 × 10^−10^
	Number of SCOPs	-	-	700	500	50
CS	RMSE (pixel)	6.6 × 10^−10^	0.58	0.29	0.29	1.7 × 10^−10^
	Time (second)	415.30	890.40	3.39	7.07	2.60
	dr_max_ (pixel)	2.4 × 10^−9^	1	0.61	0.55	3.7 × 10^−10^
	Number of SCOPs	-	-	400	200	10
BWV	RMSE (pixel)	1.2 × 10^−9^	0.56	0.29	0.28	9.8 × 10^−10^
	Time (second)	477.54	1376.80	3.57	7.97	2.41
	dr_max_ (pixel)	2.7 × 10^−9^	1	0.52	0.52	2.5 × 10^−9^
	Number of SCOPs	-	-	400	400	30
SWV	RMSE (pixel)	9.9 × 10^−10^	0.57	0.32	0.32	9.0 × 10^−10^
	Time (second)	475.13	1250.78	3.31	7.80	1.94
	dr_max_ (pixel)	2.5 × 10^−9^	1	0.72	0.71	1.7 × 10^−9^
	Number of SCOPs	-	-	500	500	10
SDWV	RMSE (pixel)	3.9 × 10^−10^	0.57	0.30	0.30	1.0 × 10^−9^
	Time (second)	490.56	1285.10	3.71	8.80	2.29
	dr_max_ (pixel)	9.3 × 10^−10^	1	0.57	0.58	1.7 × 10^−9^
	Number of SCOPs	-	-	500	500	10

## Data Availability

The satellite images to develop and evaluate the proposed method were freely downloaded from https://intelligence.airbus.com/ (Airbus-intelligence) and https://apollomapping.com/ (Apollo Mapping).

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
