# Peer review of "A Linear Regression Approach for Best Scanline Determination in the Object to Image Space Transformation Using Pushbroom Images"

_sensors, 2024, doi:10.3390/s24175594_

Round 1

Reviewer 1 Report

Comments and Suggestions for Authors

The review report

The manuscript title: A Linear Regression Approach for Best Scanline Determination in the Object to Image Space Transformation Using Pushbroom Images.

In this paper the authors present the results of using a linear regression model (LRM) to develop a new BSD framework that does not require frequent use of CE and has less computational complexity. The proposed method includes several procedural steps such as using SCOPs to calculate the unknown parameters of the LR model during the training phase and using SCPs to evaluate the accuracy of the method and its execution time during the testing phase. The proposed method was evaluated using ten different broom images, 5 million SCPs, and a limited number of SCOPs. The results yielded a root mean square error (RMSE) of the order of ten to the power of negative 9 (pixels), indicating very high accuracy. Moreover, the authors found that the proposed approach is more robust than previous known BSS/BSD methods when dealing with different broom images. The authors claim that the proposed model is suitable for practical and real-time applications.

The authors used references relevant to the topic of their article. The conclusions presented are fairly consistent with the evidence and arguments presented, and address the main question raised.

However, this article can be accepted for publication after taking into account the following points:

1. Most of references are relatively old. This research deals with modern technical topic, so the reference study should be extensive and based on modern references.

2. I think Figure 3 needs to be revised, as the LRM curves are not included in this figure.

3. The authors state in the conclusion that “The BWS method is not capable of obtaining accuracies better than 0.5 pixels and requires significant execution time.” This result is not found in the text of the manuscript!

4. Accuracy comparisons in the conclusion are qualitative and must be supported by actual numerical values ​​(in the results and conclusion).

5. The results paragraph contains only one figure (figure 3) with a brief explanation that does not include a clear comparison of numerical values. This figure must be replaced with three figures, with each figure explained separately in an appropriate manner.

6. The authors did not provide sufficient arguments to prove that the results they obtained were unique.

Author Response

Reviewer 1:

Reviewer’s comment (general): In this paper the authors present the results of using a linear regression model (LRM) to develop a new BSD framework that does not require frequent use of CE and has less computational complexity. The proposed method includes several procedural steps such as using SCOPs to calculate the unknown parameters of the LR model during the training phase and using SCPs to evaluate the accuracy of the method and its execution time during the testing phase. The proposed method was evaluated using ten different broom images, 5 million SCPs, and a limited number of SCOPs. The results yielded a root mean square error (RMSE) of the order of ten to the power of negative 9 (pixels), indicating very high accuracy. Moreover, the authors found that the proposed approach is more robust than previous known BSS/BSD methods when dealing with different broom images. The authors claim that the proposed model is suitable for practical and real-time applications.

The authors used references relevant to the topic of their article. The conclusions presented are fairly consistent with the evidence and arguments presented, and address the main question raised.

However, this article can be accepted for publication after taking into account the following points.

 Authors’ response: Thank you dear reviewer for providing valuable comments that significantly helped us improve our manuscript. Below, we have addressed all your comments. All revisions have been included in the following (you can find the yellow highlighted areas in the revised manuscript).

Reviewer’s comment (1): Most of references are relatively old. This research deals with modern technical topic, so the reference study should be extensive and based on modern references.

Authors’ response: As advised, the old references has been omitted and new ones are included in the revised manuscript.

  1. Filzmoser and K. Nordhausen, “Robust linear regression for high-dimensional data: An overview,” Wiley Interdisciplinary Reviews: Computational Statistics, vol. 13, no. 4. 2021. doi: 10.1002/wics.1524.

  1. S. Ahooei Nezhad et al., “Best Scanline Determination of Pushbroom Images for a Direct Object to Image Space Transformation Using Multilayer Perceptron,” Remote Sens., vol. 16, no. 15, p. 2787, 2024, doi: 10.3390/rs16152787.

  1. M. H. Hope, “Linear regression,” in Machine Learning: Methods and Applications to Brain Disorders, 2019, pp. 67–81. doi: 10.1016/B978-0-12-815739-8.00004-3
  2. Al-Rbaihat, H. Alahmer, A. Al-Manea, Y. Altork, M. Alrbai, and A. Alahmer, “Maximizing efficiency in solar ammonia–water absorption refrigeration cycles: Exergy analysis, concentration impact, and advanced optimization with GBRT machine learning and FHO optimizer,” Int. J. Refrig., vol. 161, pp. 31–50, 2024, doi: 10.1016/j.ijrefrig.2024.01.028
  3. P. Yee, M. S. Rusiman, S. Ismail, Suparman, F. M. Hamzah, and M. A. Shafi, “K-means clustering analysis and multiple linear regression model on household income in Malaysia,” IAES Int. J. Artif. Intell., vol. 12, no. 2, pp. 731–738, 2023, doi: 10.11591/ijai.v12.i2.pp731-738.

Reviewer’s comment (2): I think Figure 3 needs to be revised, as the LRM curves are not included in this figure.

Authors’ response: As suggested, Figure 3 has been revised. The LRM curves are shown in blue color with PM (LRM BSD) title. Because the RMSE and drmax values ​​of the NR and LRM methods are very low and close to each other, they are placed at the lowest level of the graph. Moreover, according to the fifth comment of the esteemed reviewer, Figure 3 was turned into three separate Figures. (Page 11, Section 3, Results).

Reviewer’s comment (3): The authors state in the conclusion that “The BWS method is not capable of obtaining accuracies better than 0.5 pixels and requires significant execution time.” This result is not found in the text of the manuscript!

Authors’ response: According to the results of Table 3, this method has the most required time and the most drmax among all methods. As your suggestion, the revised manuscript now includes this issue and the explanations are added to it. (Page 9, Section 3, Results)

The NR and BWS methods require 250 and 750 times more computational time than the proposed method, respectively.

On the other hand, NR and BWS require lots of CE iterations for each ground point, and their accuracy depends on this. However, the BWS cannot achieve sub-pixel accuracy, which is essential for photogrammetric applications. Since the drmax value for this method is one pixel. Additionally, BWS takes the most computation time due to the large search space and more required iteration of the CE.

Reviewer’s comment (4): Accuracy comparisons in the conclusion are qualitative and must be supported by actual numerical values ​​(in the results and conclusion).

Authors’ response: The conclusion has been revised by adding numerical values in accuracy comparison. (Page 12, Section 5, Conclusions)

The LRM method is 250 times faster than the NR method, but both methods have the same level of accuracy (in the order of ten to the power of negative 9 (pixel)). The BWS method is not capable of obtaining accuracies better than 0.5 pixels based on the obtained drmax and requires significant execution time on average of 1300 seconds. On the other hand, LRM method obtains much better accuracies than the ANN and OGP methods (RMSE values ranged between 0.28-0.32 pixels) in a shorter time and with fewer required SCOPs. The required time of these two methods is between 3 and 8 seconds and the number of required SCOPs is 500. However, the LRM method takes 2 seconds on average and the maximum number of needed SCOPs is 50.

The Results section has been updated and new explanations are also added for completeness and clarification. (Page 8, Section 3, Results)

Interestingly, no significant difference is found in the computational time between these four groups (around 2 seconds). This issue reveals that the limitation in obtaining, and the number of SCOPs does not interfere with the proposed BSD method and it achieves very high accuracy (RMSE is equal to ten to the negative power of nine) with only a small number of SCOPs (up to 50) in a very short time.

The SPI image includes urban areas with dense buildings, and this issue can be the reason for its lower accuracy compared to other images (an order of accuracy reduction in the RMSE). Furthermore, taking into account that the CS image, possessing a resolution of 6 meters, represents the lowest resolution within the used dataset, the obtained RMSE of this image reached 10 to the power of negative ten. Thus, despite using images with different characteristics, there is no correlation between the number of SCOPs, land cover, spatial resolution of the images and final accuracy.

Furthermore, the discrepancies between the RMSE and drmax of the proposed method and the NR method are negligible (Both have RMSE and drmax of the order of ten to the power of 9 or 10).

Reviewer’s comment (5): The results paragraph contains only one figure (figure 3) with a brief explanation that does not include a clear comparison of numerical values. This figure must be replaced with three figures, with each figure explained separately in an appropriate manner.

Authors’ response: As recommended, figure 3 was replaced with three figures and the descriptions of them were added in the revised manuscript. (Page 11 and 12, Section 3, Results)

According to Figure 3-5, among the compared methods, it can be observed that the NR and LRM methods exhibit superior accuracy, characterized by RMSE of ten to the negative ninth power. Subsequently, the OGP and ANN methods present RMSE of about 0.3 pixels. The BWS method demonstrates the greatest level of RMSE, with a recorded value of 0.6 pixels. Furthermore, The BWS method exhibits the most considerable computational time, amounting to 1300 seconds, due to the necessity of more repetition of the CE for each ground point across all images. The Newton-Raphson (NR) method follows closely in terms of time consumption, requiring 500 seconds, which is the second longest duration after the BWS method. This issue can also be ascribed to the iterative application of the CE; however, it is noteworthy that the frequency of repetitions for the NR method is lower than that of the BWS method. In contrast, the OGP method demonstrates an average execution time of merely 7 seconds. While both the ANN and LRM methods exhibit equivalent processing times ranging from 2 to 3 seconds. It is essential to highlight that the RMSE of the LRM is on the order of ten to the power of negative nine, whereas the RMSE of ANN method is approximately 0.3 pixels. Additionally, the behavior of the drmax can be inferred from the RMSE values. Consequently, the BWS method, characterized by the drmax, equivalent to one pixel, exhibits the maximum error in comparison to the other methods. The LRM and the NR method, both exhibiting a drmax of around ten to the power of negative nine, reside at the lower boundary of the graph, while the ANN and OGP methods occupy a mid-range position among the methodologies, achieving sub-pixel accuracy within the interval of 0.5 to 0.7 pixels. Based on this evidence, the proposed LRM method is preferable to previous studies for real-time applications due to its high accuracy, low required time, and minimum drmax.

Reviewer’s comment (6): The authors did not provide sufficient arguments to prove that the results they obtained were unique.

Authors’ response: Thank you for your insightful comment. In this paper, we introduce a new BSD approach for linear pushbroom images in a single stage. We transform 3D coordinates in object space into 2D image coordinates using BSS/BSD methods. In this study, we utilized the LRM model for this transformation. This model establishes a strong relationship between object and image spaces, making it possible to directly transform and determine the best scanlines. Our method does not rely on the Collinearity Equation (CE) or iterative search. It involves two main steps: the training phase and the testing phase. We used Simulated Control Points (SCOPs) and Simulated Check Points (SCPs) to train and test a LRM model. Initially, SCOPs were used in the LRM training step, followed by the application of SCPs to the fitted LRM model to determine the best scanline of each point without the need for CE, searching, or an iterative approach. This methodology provides sub-pixel accuracy (RMSE of the order of ten to the negative power of nine) within very short time. In the absence of accessible parameters pertaining to the EOPs of the images, these parameters may be derived utilizing the MPC model; conversely, the space resection phase is omitted. This issue emphasizes the algorithm's potential to perform independently of presumptions or obligatory parameters. Nonetheless, it is imperative to establish real GCPs in instances where the EOPs are unavailable, in addition to preparing a finite quantity of SCOPs for the training phase. Further explanations regarding this topic are provided in the Discussion and Introduction Section of the revised manuscript.

As the current solutions to solve BSS/BSD problems are still computationally demanding, time-consuming, and not feasible for real-time tasks, there is a need for faster, non-iterative, and simpler BSS/BSD methods. Since Machine Learning (ML) methods have the advantage of simplicity in implementation and higher accuracy [26], this paper pro-poses a non-iterative, innovative, and fast BSD approach by taking a new look at the regression model as one of the ML methods. This approach removes the necessity for iterative search and the use of CE in the BSD process, significantly reducing the time required. Previous methods often involved iterative procedures or the need to use CE, resulting in high time consumption in the object-to-image space transformation process. In contrast, the proposed single-step LRM method offers high speed and requires very little time. (Page 3, Section 1, Introduction).

As expected, the experiments demonstrated that the LRM-BSD method achieved sub-pixel accuracy in the shortest possible time compared to previous studies. A notable decrease in time is attributed to the omission of individual usage of CE for each ground point. In previous approaches, like NR and BWS BSS methods, the transformation of ground coordinates to image coordinates necessitated the application of CE through an iterative procedure, involving a search in the image space for each projected ground points. Conversely, the proposed method eliminates the dependence on the CE by determining the best scan-line for each ground point through a LRM regardless of searching in iterative manner. The ANN and OGP methods do not depend on CE; nevertheless, LRM-BSD is considered more advantageous due to its improved precision, efficiency, and minimum number of SCOPs requirements. The RMSE achieved by the LRM method () is suitable for photogrammetric applications, such as orthorectification with low computation time for many points. The space resection phase to obtain EOPs preceded the LRM-BSD, but if the EOPs are available from other sources, the space resection step can be omitted. Therefore, the proposed LRM-BSD approach can be used even in the absence of EOPs. This issue emphasizes the algorithm's potential to perform independently of presumptions or obligatory parameters. Nonetheless, it is imperative to establish real GCPs in instances where the EOPs are unavailable, in addition to preparing a finite quantity of SCOPs for the training phase. (Page 13, Section 4, Discussion).

Considering that the SCOPs are pivotal for the training and computation of LRM parameters, the distribution and precision of these points substantially affect the final accuracy of the proposed algorithm making use of SCPs. In the simulation phase of SCOPs and SCPs, the sole source of error arises from inaccuracies in the specification of EOPs. Given that these parameters have been derived and approximated utilizing real GCPs through the MPC model, any discrepancies in the GCPs will consequently propagate to the determination of the EOPs. Furthermore, the LRM inherently possesses a degree of error and uncertainty, which can be acknowledged as a potential source of methodological error. Nonetheless, based on the achieved RMSE and drmax, the influence of al mentioned error sources on the final accuracy could be deemed negligible. (Page 13, Section 4, Discussion).

As mentioned earlier, providing SCOPs characterized by suitable distribution across each image and precision will lead to robust accuracy of the proposed method. Keeping this point in mind regarding the inputs of the LRM method, the level of uncertainty associated with the outputs was assessed in terms of two key parameters: drmax and RMSE (overall accuracy). Based on the quantity of results derived from all images pertaining to these two parameters, the proposed method exhibits a minimal degree of uncertainty, which is deemed advantageous. Thus, due to the improved accuracy metrics presented by the LRM method, compared to previous studies, this method is considered to be more superior for application in photogrammetric contexts. (Page 13, Section 4, Discussion).

We hope the changes we made are satisfactory and we thank you again very much for your time and insightful comments.

Best regards,

Authors

Reviewer 2 Report

Comments and Suggestions for Authors

The article proposes a linear regression method that has certain innovation and practical value. The overall research process of the article is scientific and reasonable, with detailed research data. I hope the author can provide a detailed description of the literature cited and enrich references to academic literature from recent years, such as 26, 29, 31, and 37.

Comments on the Quality of English Language

High

Author Response

Reviewer 2

Reviewer’s comment (general): The article proposes a linear regression method that has certain innovation and practical value. The overall research process of the article is scientific and reasonable, with detailed research data.

Authors’ response: We thank you dear reviewer for providing valuable comments that helped us significantly improved our manuscript. Below, we have addressed your questions and incorporated your comments into the manuscript. All revisions have been included in the following (you can find the yellow highlighted areas in the revised manuscript).

Reviewer’s suggestion: I hope the author can provide a detailed description of the literature cited and enrich references to academic literature from recent years, such as 26, 29, 31, and 37.

Authors’ response: As advised, we have incorporated a brief explanation of previous studies into the introduction section of the revised manuscript (Page 2, Introduction).

In the SS method, each ground point is back-projected from the ground/object space to the image space for the total number of image scanlines, and the value of the first component of the CE is evaluated, such that if this value approaches zero or close to zero, the corresponding scanline is deemed as the best scanline. The BWS method has effectively diminished the search space of the SS approach by successively halving the image space and incorporating the first and last scanlines of the image in the assessment of the CE’s first component during an iterative procedure until the best scanline is attained. The NR method has been employed to ascertain the root of the initial component of the CE, in accordance with the inherent characteristics of linear array pushbroom images. Furthermore, this methodology necessitates the application of the CE for each ground point throughout the iterative process. The CPPS method was introduced for linear array aerial images, endeavoring to address the challenge of identifying the best scanline by delineating the CPP corresponding to each ground point through iterative process of geometric equations. Additionally, in the final stage, the CE were utilized within the iterative process in accuracy improvement stage, albeit with a reduced number of iterations. The GDP approach is based on the back-projection of each ground point to image space through CE by employing the EOPs of the first scanline. Subsequently, the GDP, which quantifies the spatial interval between the projected image point and the first line of the image, is computed and employed for updating value via specific mathematical formulations. This iterative process persists until the predetermined stopping criterion is satisfied, leading to the identification of the best scanline corresponding to the ground point using linear interpolation. This approach operates through an iterative framework involving multiple interpolations, thereby necessitating a moderate amount of computational resources. The determination of the best scanline within the framework of OGP and ANN methods is conducted through a three-step procedure. Initially, the BSS is executed by utilizing a number of simulated points, thereby computing the estimated scanline value. Subsequently, the ground points transferred from the object space to the image space using estimated scanline value, and their accuracy are refined within the image space. Following this, a model is established to relate the final scanline values calculated for the aforementioned points in conjunction with the computed approximate value. So that it can facilitate the application of this model to other points in the third step, devoid the necessity to transfer between the two spaces and independent of the CE. In the second stage, both the GA and ANN models have been employed separately.

In addition, the mentioned references are updated and new ones are included. Because reference number 29 is related to the CPP method in literature, it did not change. The new references are listed below:

  1. Filzmoser and K. Nordhausen, “Robust linear regression for high-dimensional data: An overview,” Wiley Interdisciplinary Reviews: Computational Statistics, vol. 13, no. 4. 2021. doi: 10.1002/wics.1524.

  1. S. Ahooei Nezhad et al., “Best Scanline Determination of Pushbroom Images for a Direct Object to Image Space Transformation Using Multilayer Perceptron,” Remote Sens., vol. 16, no. 15, p. 2787, 2024, doi: 10.3390/rs16152787.

  1. M. H. Hope, “Linear regression,” in Machine Learning: Methods and Applications to Brain Disorders, 2019, pp. 67–81. doi: 10.1016/B978-0-12-815739-8.00004-3
  2. Al-Rbaihat, H. Alahmer, A. Al-Manea, Y. Altork, M. Alrbai, and A. Alahmer, “Maximizing efficiency in solar ammonia–water absorption refrigeration cycles: Exergy analysis, concentration impact, and advanced optimization with GBRT machine learning and FHO optimizer,” Int. J. Refrig., vol. 161, pp. 31–50, 2024, doi: 10.1016/j.ijrefrig.2024.01.028
  3. P. Yee, M. S. Rusiman, S. Ismail, Suparman, F. M. Hamzah, and M. A. Shafi, “K-means clustering analysis and multiple linear regression model on household income in Malaysia,” IAES Int. J. Artif. Intell., vol. 12, no. 2, pp. 731–738, 2023, doi: 10.11591/ijai.v12.i2.pp731-738.

We hope the changes we made are satisfactory and we thank you again very much for your time and insightful comments.

Best regards,

Authors

Reviewer 3 Report

Comments and Suggestions for Authors

Comments to authors for “sensors-3166500-peer-review-v1” manuscript:
1. The authors need to revise the entire manuscript rigorously. The author kindly needs to make extensive efforts to adequately address all the suggested comments.
2. Manuscripts submitted to The Sensors Journal must be written in good English and proofread carefully to ensure that the research is communicated clearly.
3. It is recommended to add nomenclature, including all the abbreviations in the entire manuscript, and then put the symbols in the proper alphabetical order.
4. Authors are encouraged to include quantitative results in the abstract and conclusion sections.
5. Authors are encouraged to revise keywords.
6. Authors are encouraged to revise the method of adding references. For example, in the first paragraph of the introduction section "[1]–[5]". It is recommended to add 1 or 2 references to each single point rather than adding many references after several points.
7. The author should be able to explain the reasons behind selecting the mentioned experimental datasets. What are the most significant ones? Please be explicit when you talking about experimental datasets.
8. It is recommended to replace old references with more recent relevant ones regarding Machine Learning Approaches [2023, 2024] such as;
https://doi.org/10.1016/j.ijrefrig.2024.01.028
9. The present findings should be compared and discussed with previous works. The deviation between the current results and published data must be provided and justified.
10. The Authors must extend the analysis further considering more parametric
variations and flow physics analysis. The results Section must include more Figures (the current manuscript has only one Figure in the Results Section, which is definitely not enough). The present content is low for a journal publication.
11. A description of the proposed model and analysis, model inputs, presumptions, and operating conditions should be provided clearly.
12. The study lacks a detailed discussion of the analytical model's potential limitations, advantages, and disadvantages.
13. Model Validation analysis should be demonstrated explicitly. It is recommended to include quantitative results to identify the model's accuracy.
14. The manuscript does not address any possible sources of error in the validation process of the results with previous studies.
15. Insufficient exploration of potential uncertainties or assumptions in the study.

Comments on the Quality of English Language

 Minor editing of English language required.

Author Response

Reviewer 1:

Reviewer’s comment (general): The authors need to revise the entire manuscript rigorously. The author kindly needs to make extensive efforts to adequately address all the suggested comments. Manuscripts submitted to The Sensors Journal must be written in good English and proofread carefully to ensure that the research is communicated clearly.

Authors’ response: We thank you dear reviewer for providing valuable comments that significantly helped us improve our manuscript. Below, we have addressed your questions and incorporated your comments into the manuscript. All revisions have been included in the following (you can find the yellow highlighted areas in the revised manuscript).

Reviewer’s comment (1): It is recommended to add nomenclature, including all the abbreviations in the entire manuscript, and then put the symbols in the proper alphabetical order.

Authors’ response: As recommended, a nomenclature was added in the revised manuscript before References list (Page 13, Nomenclature).

Abbreviations

Full description

ANN

Artificial Neural Network

AP

image of Annapolis, Pleiades

AS

image of Amsterdam, SPOT

BSD

Best Scanline Determination

BSS

Best Scanline Search

BWS

Bisecting Window Search

BWV

image of Boulder, WorldView

CE

Collinearity Equation

CPP

Central Perspective Plane

CS

image of Curitiba, SPOT

DTM

Digital Terrain Model

DEM

Digital Elevation Model

EOP

Exterior Orientation Parameters

GA

Genetic Algorithm

GCP

Ground Control Points

GDP

General Distance Prediction

IOP

Interior Orientation Parameters

JQB

image of Jaipur, QuickBird

JS

image of Jaicos, SPOT

LR

Linear Regression

LRM

Linear Regression Model

LSM

Least Square Method

MLR

Multivariate Linear Regression

MP

image of Melbourne, Pleiades

MPC

Multiple Projection Center

NR

Newton Raphson

OGP

Optimal Global Polynomial

PR

Polynomial Regression

PRM

Polynomial Regression Model

RAM

Random Access Memory

RMSE

Root Mean Square Error

RPC

Rational Polynomial Coefficients

SCOP

Simulated Control Points

SCP

Simulated Check Points

SDWV

image of SanDiego,WorldView

SLR

Simple Linear Regression

SPI

image of SaoPaulo, IKONOS

SS

Sequential Search

SWV

image of Sydney, WorldView

Reviewer’s comment (2): Authors are encouraged to include quantitative results in the abstract and conclusion sections.

Authors’ response: As suggested, the revised manuscript now contains the quantitative results in the abstract and conclusion. (Page 1, Abstract and Page 12, Section 5, Conclusions).

The Root Mean Square Error (RMSE) was found to be in the order of ten to the power of negative nine (pixel), indicating very high accuracy. Furthermore, the proposed approach is more robust than previous well-known BSS/BSD methods when handling various pushbroom images, making it suitable for practical and real-time applications due to its high speed, which only requires 2-3 seconds of time.

The LRM method is 250 times faster than the NR method, but both methods have the same level of accuracy (in the order of ten to the power of negative nine (pixel)). The BWS method is not capable of obtaining accuracies better than 0.5 pixels based on the obtained drmax and requires significant execution time on average of 1300 seconds. On the other hand, LRM method obtains much better accuracies than the ANN and OGP methods (RMSE values ranged between 0.28-0.32 pixels) in a shorter time and with fewer required SCOPs. The required time of these two methods is between 3 and 8 seconds and the number of required SCOPs is 500. However, the LRM method takes 2 seconds on average and the maximum number of needed SCOPs is 50.

Reviewer’s comment (3):  Authors are encouraged to revise keywords.

Authors’ response: As suggested, the keywords are modified and included in the revised manuscript.

Keywords: Object-to-Image Transformation; Best Scanline Search (BSS); Best Scanline Determination (BSD); Linear Regression Model (LRM); Machine Learning (ML); Collinearity Equation (CE); Photogrammetry

Reviewer’s comment (4): Authors are encouraged to revise the method of adding references. For example, in the first paragraph of the introduction section "[1]–[5]". It is recommended to add 1 or 2 references to each single point rather than adding many references after several points.

Authors’ response: According to your suggestion, the style of referencing has been revised through the whole manuscript. The changes are highlighted in the revised manuscript.

Reviewer’s comment (5): The author should be able to explain the reasons behind selecting the mentioned experimental datasets. What are the most significant ones? Please be explicit when you talking about experimental datasets.

Authors’ response: The details of the experimental datasets were added in the revised manuscript (Page 4, Section 2.1, Dataset Description).

As reported in Table.1, the experimental datasets include ten different satellite pushbroom images acquired by various sensors including IKONOS, Pleiades 1A and 1B, QuickBird, SPOT6, SPOT7, WorldView 1, and WorldView 2. A comprehensive selection of ten images was made, illustrating a variety of sensor characteristics, including dimensions and spatial resolutions, alongside disparate land cover and topographical conditions, such as urban areas, flat terrains, agricultural regions, and combinations thereof, for a more thorough assessment of the proposed BSD method (see Figure 2). Specifically, the spatial resolution of these images exhibited a range from 0.5 to 6 meters. It is noteworthy that all images utilized in this research were supplemented by Rational Polynomial Coefficients (RPC) files; the necessary Ground Control Points (GCPs) were derived from these supplementary files. Furthermore, elevation data for the GCPs as well as the average height of the study areas were derived from the accessible Digital Elevation Model (DEM) resources.

Reviewer’s comment (6):  It is recommended to replace old references with more recent relevant ones regarding Machine Learning Approaches [2023, 2024] such as;

https://doi.org/10.1016/j.ijrefrig.2024.01.028

Authors’ response: As recommended, the old references are replaced with more recent ones, which are also listed below. The mentioned reference is used in Page 3, Section 1, Introduction.

Since Machine Learning (ML) methods have the advantage of simplicity in implementation and higher accuracy [26], this paper proposes a non-iterative, innovative, and fast BSD approach by taking a new look at the regression model as one of the ML methods.

  1. Filzmoser and K. Nordhausen, “Robust linear regression for high-dimensional data: An overview,” Wiley Interdisciplinary Reviews: Computational Statistics, vol. 13, no. 4. 2021. doi: 10.1002/wics.1524.

  1. S. Ahooei Nezhad et al., “Best Scanline Determination of Pushbroom Images for a Direct Object to Image Space Transformation Using Multilayer Perceptron,” Remote Sens., vol. 16, no. 15, p. 2787, 2024, doi: 10.3390/rs16152787.

  1. M. H. Hope, “Linear regression,” in Machine Learning: Methods and Applications to Brain Disorders, 2019, pp. 67–81. doi: 10.1016/B978-0-12-815739-8.00004-3
  2. Al-Rbaihat, H. Alahmer, A. Al-Manea, Y. Altork, M. Alrbai, and A. Alahmer, “Maximizing efficiency in solar ammonia–water absorption refrigeration cycles: Exergy analysis, concentration impact, and advanced optimization with GBRT machine learning and FHO optimizer,” Int. J. Refrig., vol. 161, pp. 31–50, 2024, doi: 10.1016/j.ijrefrig.2024.01.028
  3. P. Yee, M. S. Rusiman, S. Ismail, Suparman, F. M. Hamzah, and M. A. Shafi, “K-means clustering analysis and multiple linear regression model on household income in Malaysia,” IAES Int. J. Artif. Intell., vol. 12, no. 2, pp. 731–738, 2023, doi: 10.11591/ijai.v12.i2.pp731-738.

Reviewer’s comment (7): The present findings should be compared and discussed with previous works. The deviation between the current results and published data must be provided and justified.

Authors’ response: As recommended, the revised manuscript now provides the further descriptions of the present finding in comparison of previous studies.

The proposed method requires significantly less time than all the other methods. The NR and BWS methods require 250 and 750 times more computational time than the pro-posed method, respectively. The reason for this significant reduction in time is the non-iterative procedure used in the proposed method, regardless of using the CE. On the other hand, the NR and BWS methods require lots of CE iterations for each ground point, and their accuracy depends on this. However, the BWS method cannot achieve sub-pixel accuracy, which is essential for photogrammetric applications. Since the drmax value for this method is one pixel. Additionally, the BWS method takes the most computation time due to the large search space and more required iteration of the CE.

Furthermore, the discrepancies between the RMSE and drmax of the proposed method and the NR method are negligible (Both have RMSE and drmax of the order of ten to the power of 9 or 10). The comparison of the ANN and OGP BSD methods with the proposed method reveals that the LRM method achieves higher RMSE by using fewer SCOPs in less time. The main reason for this difference is that the ANN and OGP BSD methods require more SCOPs than the LRM method in the error modeling phase. The number and distribution of these SCOPs directly affect the final accuracy of modeling and computation time to ensure the accuracy of determining the best scanline. (Page 10, Section 3, Results)

According to Figure 3-5, among the compared methods, it can be observed that the NR and LRM methods exhibit superior accuracy, characterized by RMSE of ten to the negative ninth power. Subsequently, the OGP and ANN methods present RMSE of about 0.3 pixels. The BWS method demonstrates the greatest level of RMSE, with a recorded value of 0.6 pixels. Furthermore, The BWS method exhibits the most considerable computational time, amounting to 1300 seconds, due to the necessity of more repetition of the CE for each ground point across all images. The Newton-Raphson (NR) method follows closely in terms of time consumption, requiring 500 seconds, which is the second longest duration after the BWS method. This issue can also be ascribed to the iterative application of the CE; however, it is noteworthy that the frequency of repetitions for the NR method is lower than that of the BWS method. In contrast, the OGP method demonstrates an average execution time of merely 7 seconds. While both the ANN and LRM methods exhibit equivalent processing times ranging from 2 to 3 seconds. It is essential to highlight that the RMSE of the LRM is approximately ten to the power of negative nine, whereas the RMSE of ANN method is approximately 0.3 pixels. Additionally, the behavior of the drmax can be inferred from the RMSE values. Consequently, the BWS method, characterized by the drmax, equivalent to one pixel, exhibits the maximum error in comparison to the other methods. The LRM and the NR method, both exhibiting a drmax of around ten to the power of negative nine, reside at the lower boundary of the graph, while the ANN and OGP methods occupy a mid-range position among the methodologies, achieving sub-pixel accuracy within the interval of 0.5 to 0.7 pixels. Based on this evidence, the proposed LRM method is preferable to previous studies for real-time applications due to its high accuracy, low required time, and minimum drmax. (Page 11 and 12, Section 3, Results)

Reviewer’s comment (8): The Authors must extend the analysis further considering more parametric variations and flow physics analysis. The results Section must include more Figures (the current manuscript has only one Figure in the Results Section, which is definitely not enough). The present content is low for a journal publication.

Authors’ response: The number of Figures are increased in Results Section. The additional details are included in the Results section for further clarity and completeness.

The SPI image includes urban areas with dense buildings, and this issue can be the reason for its lower accuracy compared to other images (an order of accuracy reduction in the RMSE). Furthermore, taking into account that the CS image, possessing a resolution of 6 meters, represents the lowest resolution within the used dataset, the obtained RMSE of this image reached 10 to the power of negative ten. Thus, despite using images with different characteristics, there is no correlation between the number of SCOPs, land cover, spatial resolution of the images and final accuracy. (Page 9, Section 3, Results)

The NR and BWS methods require 250 and 750 times more computational time than the proposed method, respectively. The reason for this significant reduction in time is the non-iterative procedure used in the proposed method, regardless of using the CE. On the other hand, the NR and BWS methods require lots of CE iterations for each ground point, and their accuracy depends on this. However, the BWS method cannot achieve sub-pixel accuracy, which is essential for photogrammetric applications. Since the drmax value for this method is one pixel. Additionally, the BWS method takes the most computation time due to the large search space and more required iteration of the CE.

Furthermore, the discrepancies between the RMSE and drmax of the proposed method and the NR method are negligible (Both have RMSE and drmax of the order of ten to the power of 9 or 10). (Page 10, Section 3, Results)

According to Figure 3-5, among the compared methods, it can be observed that the NR and LRM methods exhibit superior accuracy, characterized by RMSE of ten to the negative ninth power. Subsequently, the OGP and ANN methods present RMSE of about 0.3 pixels. The BWS method demonstrates the greatest level of RMSE, with a recorded value of 0.6 pixels. Furthermore, The BWS method exhibits the most considerable computational time, amounting to 1300 seconds, due to the necessity of more repetition of the CE for each ground point across all images. The Newton-Raphson (NR) method follows closely in terms of time consumption, requiring 500 seconds, which is the second longest duration after the BWS method. This issue can also be ascribed to the iterative application of the CE; however, it is noteworthy that the frequency of repetitions for the NR method is lower than that of the BWS method. In contrast, the OGP method demonstrates an average execution time of merely 7 seconds. While both the ANN and LRM methods exhibit equivalent processing times ranging from 2 to 3 seconds. It is essential to highlight that the RMSE of the LRM is on the order of ten to the power of negative nine, whereas the RMSE of ANN method is approximately 0.3 pixels. Additionally, the behavior of the drmax can be inferred from the RMSE values. Consequently, the BWS method, characterized by the drmax, equivalent to one pixel, exhibits the maximum error in comparison to the other methods. The LRM and the NR method, both exhibiting a drmax of around ten to the power of negative nine, reside at the lower boundary of the graph, while the ANN and OGP methods occupy a mid-range position among the methodologies, achieving sub-pixel accuracy within the interval of 0.5 to 0.7 pixels. Based on this evidence, the proposed LRM method is preferable to previous studies for real-time applications due to its high accuracy, low required time, and minimum drmax. (Page 11 and 12, Section 3, Results)

Reviewer’s comment (9): A description of the proposed model and analysis, model inputs, presumptions, and operating conditions should be provided clearly.

Authors’ response: A new table containing the inputs and outputs of each step has been added for better comprehension and detailed explanation of the proposed method (Page 4, Section 2, Table 1).

An overview of the proposed method is given in Table.1 for clarification.

Table 1. An overview of the proposed method

Mathematical Model

Inputs

Outputs

Preprocessing steps

Space resection

·       MPC model

·       IOPs

·       Image coordinates of GCPs

·       Ground coordinates of GCPs

·       EOPs of the scanlines for the whole image

SCOPs and SCPs generation

·       The CE

·       IOPs

·       EOPs

·       Mean height of the study area

·       Image and ground coordinates of SCOPs

·       Image and ground coordinates of SCPs

LRM BSD steps

Training phase

·       LRM model

·       Image coordinates of SCOPs

·       Ground coordinates of SCOPs

·       Fitted LRM model for object-to-image transformation

Testing phase

·       LRM model

·       Ground coordinates of SCPs

·       Image coordinates of SCPs predicted through the fitted LRM model

·       Accuracy assessment (RMSE, time, drmax)

Reviewer’s comment (10): The study lacks a detailed discussion of the analytical model's potential limitations, advantages, and disadvantages.

Authors’ response: In this paper, we introduce a new BSD approach for linear pushbroom images in a single stage. We transform 3D coordinates in object space into 2D image coordinates using BSS/BSD methods. In this study, we utilized the LRM model for this transformation. This model establishes a strong relationship between object and image spaces, making it possible to directly transform and determine the best scanlines. Our method does not rely on the Collinearity Equation (CE) or iterative search. It involves two main steps: the training phase and the testing phase. We used Simulated Control Points (SCOPs) and Simulated Check Points (SCPs) to train and test a LRM model. Initially, SCOPs were used in the LRM training step, followed by the application of SCPs to the fitted LRM model to determine the best scanline of each point without the need for CE, searching, or an iterative approach. This methodology provides sub-pixel accuracy (RMSE of the order ten to the negative power of nine) within very short time. In the absence of accessible parameters pertaining to the EOPs of the images, these parameters may be derived utilizing the MPC model; conversely, the space resection phase is omitted. This issue emphasizes the algorithm's potential to perform independently of presumptions or obligatory parameters. Nonetheless, it is imperative to establish real GCPs in instances where the EOPs are unavailable, in addition to preparing a finite quantity of SCOPs for the training phase. Further explanations regarding this topic are provided in the Discussion section of the revised manuscript.

As expected, the experiments demonstrated that the LRM-BSD method achieved sub-pixel accuracy in the shortest possible time compared to previous studies. A notable decrease in time is attributed to the omission of individual usage of CE for each ground point. In previous approaches, like NR and BWS BSS methods, the transformation of ground coordinates to image coordinates necessitated the application of CE through an iterative procedure, involving a search in the image space for each projected ground points. Conversely, the proposed method eliminates the dependence on the CE by determining the best scan-line for each ground point through a LRM regardless of searching in iterative manner. The ANN and OGP methods do not depend on CE; nevertheless, LRM-BSD is considered more advantageous due to its improved precision, efficiency, and minimum number of SCOPs requirements. The RMSE achieved by the LRM method () is suitable for photogrammetric applications, such as orthorectification with low computation time for many points. The space resection phase to obtain EOPs preceded the LRM-BSD, but if the EOPs are available from other sources, the space resection step can be omitted. Therefore, the proposed LRM-BSD approach can be used even in the absence of EOPs. This issue emphasizes the algorithm's potential to perform independently of presumptions or obligatory parameters. Nonetheless, it is imperative to establish real GCPs in instances where the EOPs are unavailable, in addition to preparing a finite quantity of SCOPs for the training phase. (Page 13 and 14, Section 4, Discussion).

Reviewer’s comment (11): Model Validation analysis should be demonstrated explicitly. It is recommended to include quantitative results to identify the model's accuracy.

Authors’ response: Quantitative results are included in the Results Section of the revised manuscript.

The SPI image includes urban areas with dense buildings, and this issue can be the reason for its lower accuracy compared to other images (an order of accuracy reduction in the RMSE). Furthermore, taking into account that the CS image, possessing a resolution of 6 meters, represents the lowest resolution within the used dataset, the obtained RMSE of this image reached 10 to the power of negative ten. Thus, despite using images with different characteristics, there is no correlation between the number of SCOPs, land cover, spatial resolution of the images and final accuracy. (Page 9, Section 3, Results)

The NR and BWS methods require 250 and 750 times more computational time than the proposed method, respectively. The reason for this significant reduction in time is the non-iterative procedure used in the proposed method, regardless of using the CE. On the other hand, the NR and BWS methods require lots of CE iterations for each ground point, and their accuracy depends on this. However, the BWS method cannot achieve sub-pixel accuracy, which is essential for photogrammetric applications. Since the drmax value for this method is one pixel. Additionally, the BWS method takes the most computation time due to the large search space and more required iteration of the CE.

Furthermore, the discrepancies between the RMSE and drmax of the proposed method and the NR method are negligible (Both have RMSE and drmax of the order of ten to the power of 9 or 10). (Page 10, Section 3, Results)

According to Figure 3-5, among the compared methods, it can be observed that the NR and LRM methods exhibit superior accuracy, characterized by RMSE of ten to the negative ninth power. Subsequently, the OGP and ANN methods present RMSE of about 0.3 pixels. The BWS method demonstrates the greatest level of RMSE, with a recorded value of 0.6 pixels. Furthermore, The BWS method exhibits the most considerable computational time, amounting to 1300 seconds, due to the necessity of more repetition of the CE for each ground point across all images. The Newton-Raphson (NR) method follows closely in terms of time consumption, requiring 500 seconds, which is the second longest duration after the BWS method. This issue can also be ascribed to the iterative application of the CE; however, it is noteworthy that the frequency of repetitions for the NR method is lower than that of the BWS method. In contrast, the OGP method demonstrates an average execution time of merely 7 seconds. While both the ANN and LRM methods exhibit equivalent processing times ranging from 2 to 3 seconds. It is essential to highlight that the RMSE of the LRM is on the order of ten to the power of negative nine, whereas the RMSE of ANN method is approximately 0.3 pixels. Additionally, the behavior of the drmax can be inferred from the RMSE values. Consequently, the BWS method, characterized by the drmax, equivalent to one pixel, exhibits the maximum error in comparison to the other methods. The LRM and the NR method, both exhibiting a drmax of around ten to the power of negative nine, reside at the lower boundary of the graph, while the ANN and OGP methods occupy a mid-range position among the methodologies, achieving sub-pixel accuracy within the interval of 0.5 to 0.7 pixels. Based on this evidence, the proposed LRM method is preferable to previous studies for real-time applications due to its high accuracy, low required time, and minimum drmax. (Page 11 and 12, Section 3, Results)

Reviewer’s comment (12): The manuscript does not address any possible sources of error in the validation process of the results with previous studies.

Authors’ response: As suggested, the mentioned issue was included in the revised manuscript. (Page 13, Section 4, Discussion).

Considering that the SCOPs are pivotal for the training and computation of LRM parameters, the distribution and precision of these points substantially affect the final accuracy of the proposed algorithm making use of SCPs. In the simulation phase of SCOPs and SCPs, the sole source of error arises from inaccuracies in the specification of EOPs. Given that these parameters have been derived and approximated utilizing real GCPs through the MPC model, any discrepancies in the GCPs will consequently propagate to the determination of the EOPs. Furthermore, the LRM inherently possesses a degree of error and uncertainty, which can be acknowledged as a potential source of methodological error. Nonetheless, based on the achieved RMSE and drmax, the influence of al mentioned error sources on the final accuracy could be deemed negligible.

Reviewer’s comment (13): Insufficient exploration of potential uncertainties or assumptions in the study.

Authors’ response: The relevant explanations of the mentioned issues are included in the Introduction and Discussion Section of the revised manuscript.

As the current solutions to solve BSS/BSD problems are still computationally demanding, time-consuming, and not feasible for real-time tasks, there is a need for faster, non-iterative, and simpler BSS/BSD methods. Since Machine Learning (ML) methods have the advantage of simplicity in implementation and higher accuracy [26], this paper pro-poses a non-iterative, innovative, and fast BSD approach by taking a new look at the regression model as one of the ML methods. This approach removes the necessity for iterative search and the use of CE in the BSD process, significantly reducing the time required. Previous methods often involved iterative procedures or the need to use CE, resulting in high time consumption in the object-to-image space transformation process. In contrast, the proposed single-step LRM method offers high speed and requires very little time. (Page 3, Section 1, Introduction).

In addition to the explanations of the previous comment that was added in the Discussion Section, the following sentences were also included for completeness and clarification.

As mentioned earlier, providing SCOPs characterized by suitable distribution across each image and precision will lead to robust accuracy of the proposed method. Keeping this point in mind regarding the inputs of the LRM method, the level of uncertainty associated with the outputs was assessed in terms of two key parameters: drmax and RMSE (overall accuracy). Based on the quantity of results derived from all images pertaining to these two parameters, the proposed method exhibits a minimal degree of uncertainty, which is deemed advantageous. Thus, due to the improved accuracy metrics presented by the LRM method, compared to previous studies, this method is considered to be more superior for application in photogrammetric contexts.

We hope the changes we made are satisfactory and we thank you again very much for your time and insightful comments.

Best regards,

Authors

Round 2

Reviewer 1 Report

Comments and Suggestions for Authors

The authors have completed the required revisions, so I suggest accepting the article for publication.

Reviewer 3 Report

Comments and Suggestions for Authors

The comments are satisfied.